# Acute Toxicity of the Recombinant and Native Phα1β Toxin: New Analgesic from *Phoneutria nigriventer* Spider Venom

**DOI:** 10.3390/toxins10120531

**Published:** 2018-12-12

**Authors:** Eliane Dallegrave, Eliane Taschetto, Mirna Bainy Leal, Flavia Tasmim Techera Antunes, Marcus Vinicius Gomez, Alessandra Hubner de Souza

**Affiliations:** 1Department of Pharmacosciences, Federal University of Health Sciences of Porto Alegre, Porto Alegre, RS 90050-170 Brazil; elianedal@ufcspa.edu.br; 2Postgraduate Program in Genetics and Applied Toxicology, Lutheran University of Brazil, Canoas, RS 92425–900, Brazil; etaschetto@gmail.com; 3Department of Pharmacology, Institute of Basic Health Sciences, Universidade Federal do Rio Grande do Sul, Porto Alegre, RS 90050-17, Brazil; mirnabl@gmail.com; 4Postgraduate Program in Cellular and Molecular Biology Applied of Health, Lutheran University of Brazil, Canoas, RS 92425–900, Brazil; tasmin@hotmail.com; 5Postgraduate Program in Health Sciences: Medicine and Biomedicine, Institute of Education and Research, Grupo Santa Casa de Belo Horizonte, Belo Horizonte, MG 30150-240, Brazil; marcusvgomez@gmail.com

**Keywords:** *Phoneutria nigriventer*, Phα1β, acute toxicity

## Abstract

Phα1β, a purified peptide from the venom of the spider *Phoneutria nigriventer*, and its recombinant form CTK 01512-2 are voltage-dependent calcium channel (Ca_V_) blockers of types N, R, *P*/Q, and L with a preference for type N. These peptides show analgesic action in different pain models in rats. The aim of this study was to evaluate the acute intrathecal toxicity of the native and recombinant Phα1β toxin in Wistar rats. Clinical signs, serum biochemistry, organ weight, and histopathological alterations were evaluated in male and/or female rats. Dyspnea was observed in males, hyporesponsiveness in females, and Straub tail and tremors in both genders. There were no significant differences in male organ weight, although significant differences in the female relative weight of the adrenal glands and spleen have been observed; these values are within the normal range. Serum biochemical data revealed a significant reduction within the physiological limits of species related to urea, ALT, AST, and FA. Hepatic and renal congestion were observed for toxin groups. In renal tissue, glomerular infiltrates were observed with increased glomerular space. These histological alterations were presented in focal areas and in mild degree. Therefore, Phα1β and CTK 01512-2 presented a good safety profile with transient toxicity clinical signals in doses higher than used to obtain the analgesic effect.

## 1. Introduction

*Phoneutria nigriventer* venom is studied for containing low molecular weight non-protein compounds and approximately 41 neurotoxins with the potential capacity to cross the blood-brain barrier [1,2,3]. PhTx3 is the most studied fraction; six toxins have been characterized and their peptides act on voltage-dependent calcium (Ca_V_) channels. The PnTx3-6 toxin (also known as Phα1β) inhibits Ca_V_ channels of type N, R, and *P*/Q, and has been pre-clinically proven to have an antinociceptive role in several pain models [4,5,6,7], and CTK 01512-2, the recombinant form of Phα1β, shows similar activity [8,9,10,11,12]. Its mechanism of action is mainly based on the blockade of Ca_V_ 2.2 (type N), decreasing the release of the neurotransmitter glutamate in the dorsal horn of the spinal cord [4,13].

The ω-conotoxin MVIIA, derived from the venom of *Conus magus*, marketed by the name Prialt^®^ (ziconotide), has a pharmacological role similar to the spider toxin of this study. It has been approved for clinical use by the Food and Drug Administration in the United States since 2004 and by the European Medicine Agency in 2005 indicated for chronic pain management in adults refractory to other treatments, although it has serious adverse effects [14,15,16,17].

Considering the need to develop potent and safe analgesics for use in severe and disabling pain and meeting the requirements of the Organization for Economic Cooperation and Development (OECD) Test Guidelines (based on guideline 420/2001), this study aimed to evaluate the safety of the intrathecal acute exposure to Phα1β native toxin and CTK 1512-2 compared to ω-conotoxin (MVIIA), using an animal model. This information is needed to meet pre-clinical research criteria by subsidizing the protection of human health and helping to select an appropriate starting dose.

## 2. Results

### 2.1. Male Protocols

#### 2.1.1. Clinical Sings

The acute effects of the intrathecal administration of CTK 01512-2 and native Phα1β toxin, MVIIA, and PBS in male rats are presented in Table 1. Clinical signs manifested include an increase or decrease of the ambulation, hypnosis, decrease to touch response, grooming, piloerection, ataxia, dyspnea, tremors, and Straub tail.

Initially, all males showed an increase in ambulation. After 15 min, a decrease in ambulation accompanied by hypnosis in most of the animals was observed. Hypnosis was not observed in the native toxin group, differing significantly from the other groups (*p* = 0.043, chi-square). Grooming was observed in most control animals, but only in the first hour. Experimental groups showed grooming for a longer period, in some cases, by all animals and during the whole observation period. Piloerection was observed in all groups, reaching the totality of the animals exposed to the native toxin, throughout the period. Only a few animals showed a decrease in response to touch at different times and no difference between the groups was observed. Ataxia and tremors were observed only in an animal treated with the MVIIA. Dyspnea was manifested in all experimental groups, differing significantly from controls (*p* = 0.003, chi-square). Straub tail was observed in only two males, one being treated with the native toxin and the other with the MVIIA.

#### 2.1.2. Relative Organ Weight

Table 2 shows data on the relative weight of male organs after 24 h of treatment. There was no significant difference between the groups in the relative adrenal gland, lung, and spleen (*p* = 0.989, *p* = 0.593, *p* = 0.353, respectively, one-way ANOVA). Although the relative weight of the other organs did not present normal distribution, the non-parametric analysis (Kruskal–Wallis) revealed that there was no significant difference between the groups in the relative weight of the liver, kidneys, and heart (*p* = 0.608, *p* = 0.568, *p* = 0.809, respectively).

#### 2.1.3. Biochemical Parameters

The biochemical parameters evaluated 24 h after intrathecal treatment are shown in Table 3. There was no significant difference between groups in creatinine (*p* = 0.170; one-way ANOVA), although there was a significant difference in urea (*p* = 0.004; one-way ANOVA), which was lower in the intermediate and high recombinant doses and the native toxin groups than others. AST, ALT, ALP, Amylase, and CK did not show normal distribution and were compared by Kruskal–Wallis non-parametric test. There was no significant difference between the groups regarding amylase and CK (*p* = 0.250, *p* = 0.209, respectively). However, there was a significant difference between AST, ALT, and AL*P* groups (*p* = 0.034, *p* = 0.034, and *p* = 0.039, respectively). AST was significantly lower in the groups treated with the intermediate dose of the recombinant and native toxin compared to the others, except in the group treated with a high dose of the recombinant toxin. In relation to ALP, only native Phα1β toxin and MVIIA showed lower values than the other groups.

#### 2.1.4. Histopathology

The histopathologic alterations observed 24 h after acute intrathecal administration of CTK 01512-2 and native Phα1β toxin, MVIIA, and PBS in male rats are presented in Figure 1 (liver) and Figure 2 (kidneys). Although the hepatic tissue structure was not altered, some cellular changes were observed. In the MVIIA group (Figure 1B), hydropic degeneration (20%), vacuolization (100%), congestion (100%), and inflammatory infiltrate (20%) were observed, similarly to native Phα1β toxin-treated animals (Figure 1F). Vacuolization (80%), congestion (100%), and inflammatory infiltrate (20%) were observed in the CTK 01512-2 Phα1β toxin groups but only for animals treated with intermediate (Figure 1D) and high (Figure 1E) doses. The animals treated with the low (Figure 1C) dose only showed vacuolization (60%) and congestion (100%). Different from all toxin groups, the control (Figure 1A) did not manifest these alterations. Statistical significance was observed only for hepatic cells vacuolization and congestion related to the control group (*p* < 0.05, chi-square).

Histological analysis of renal tissue revealed hydropic degeneration and congestion in all animals of the CTK 01512-2 Phα1β toxin groups (100%) (Figure 2C-low, D-intermediate, and E-high dose) and inflammatory infiltrate and glomerular space increase in the Bowman’s capsule in MVIIA (Figure 2B) and native Phα1β toxin (60%) (Figure 2F) groups. Statistically significant difference was observed for all recombinant toxin groups in renal cells vacuolization and congestion (*p* < 0.05, chi-square), and MVIIA and native Phα1β toxin were different from the other groups (*p* < 0.05, chi-square) in terms of inflammatory infiltrate and glomerular space increase. Although histopathological changes were observed in many animals, these were presented in focal areas and to mild degrees.

### 2.2. Female Protocols

#### 2.2.1. Clinical Sings

In females, the clinical signs manifested after acute intrathecal administration of CTK01512-2 and native Phα1β toxin, MVIIA, and PBS included an increase or decrease of the ambulation, hypnosis, decrease to touch response, grooming, piloerection, dyspnea, tremors, and Straub tail (Table 4). Like the behavior manifested by males, the females initially presented an increase in ambulation. After 15 min, a decrease in ambulation was observed in all animals but the hypnosis was only observed in some females. Although it did not show a statistically significant difference versus male rats, the females of the native toxin group did not show hypnosis either. Grooming was observed in most animals, but at different times. Piloerection and dyspnea were observed in all groups, but in control females for a shorter time period. All treatment groups showed females with decreased touch response; however, this behavior was significantly higher (*p* = 0.015, chi-square) in the animals treated with MVIIA, an intermediate dose of CTK 01512-2, or native Phα1β toxin. Tremors were manifested in all experimental groups, differing significantly from controls (*p* = 0.031, chi-square). Straub tail was observed in two females treated with MVIIA and one female treated with a high dose of CTK 01512-2.

#### 2.2.2. Bodyweight

The relative body weight did not present a statistically significant difference between groups (*p* = 0.363, Repeated Measures ANOVA) in the post-treatment period of 14 days (Figure 3).

#### 2.2.3. Relative Organ Weight

Table 5 shows data on the relative weight of female organs after 14 days of treatment. There was no significant difference between the groups in the relative liver, kidney, heart, and lung weight (*p* = 0.060, *p* = 0.877, *p* = 0.782, *p* = 0.774, respectively, one-way ANOVA). There was a significant difference between the groups in the relative adrenal and spleen weight (*p* = 0.0001; *p* = 0.022, respectively, one-way ANOVA). Adrenals from the groups treated with 500 pmol/site of the recombinant and native toxin were significantly higher than adrenals from the group treated with 200 pmol/site of the recombinant toxin. The spleen of the native toxin treated group was significantly smaller than the control group treated with PBS. Although significant differences in the relative weight of some organs have been observed, they have no biological significance considering that these values are similar to the historical control and no macroscopic alterations were observed at necropsy.

## 3. Discussion

The present study verified the acute intrathecal toxicity of the native Phα1β and CTKα1515–2 toxins compared to another analgesic toxin, the ω-conotoxin MVIIA. Significant preclinical signs were found in MVIIA, native Phα1β, and at high doses of CTK 01512-2. Straub’s tail was found in MVIIA (males and females). T3 and native Phα1β revealed the occurrence of strong spasms in the musculature of the base of the tail, indicative of a significant reaction in the CNS [18] as also found in other studies of MVIIA and native Phα1β [4].

Piloerection was verified in males of all groups and was more pronounced in the MVIIA and Phα1β groups, which shows alteration in the autonomous nervous system. Ataxia or difficulty walking and inability to stand up, accompanied by weight loss and dyspnea, was found in MVIIA, T3, and native Phα1β groups, indicating systemic alteration. All treatments showed tremors in females, but there were no significant changes in body weight over the fourteen days. These data corroborate the findings of other studies that found that MVIIA produces adverse effects at analgesic doses, such as sleepiness, sedation, motor dysfunction, and paradoxical hyperalgesia [9,19].

In both genders, touch response was decreased, with palpebral ptosis and hypnosis at varying levels, indicating a depressant effect in the central nervous system. However, hyperexcitability was detected in animals treated with Phα1β and MVIIA. The origin of the side effects of Ca_V_ 2.2 inhibitory peptides is unknown, but the toxicity of MVIIA should mainly arise from a methionine residue inserted into a hydrophobic hole located between repeats II and III of Ca_V_ 2.2 [20]. It is known that other ω-conotoxins have a higher therapeutic index than MVIIA in models of neuropathic pain, but the hypothesis of faster reversibility of the binding linked to the difference in the peptide sequence is discarded [3,21].

The toxin target of this work has 55 amino acids, of which 12 residues are cysteines with 6 disulfide bridges, but it is possible that it exists in different forms containing different disulfide bridge arrays, while maintaining the same function and activity [22]. Already MVIIA contains 25 amino acids, of which 6 are cysteine residues linked with 3 disulfide bonds [23]. These biophysical properties may lead to changes in pharmacokinetic profiles of toxins in the organic system [24], which will be the subject of future research since CTK 01512-2 has already been described in terms of its effectiveness in both intrathecal and intravenous administration [8].

The analysis of organ weight showed no significant changes in males. Although in females, the adrenal glands and spleen presented a significant decrease, this change was inside normal parameters for the species. Similarly, regarding the biochemical parameters, it is suggested that the studied toxins could be safe, since the values found for urea, AST, ALT, and ALP were reduced compared to the control group and within the physiologic species limits, without clinical significance [25].

In males, the histopathological changes, including hepatic congestion and cell vacuolization as well as inflammatory infiltrate and increase in Bowman’s space, can be due to an acute inflammatory reaction induced by intrathecal administration of the all toxins, in the same way as the reaction observed to *Phoneutria* spider venom [26].

Although all toxins showed slight and focal histopathological alterations in liver and kidneys, the effects observed are probably reversible, considering that in females the body weight and clinical signs fourteen days after treatment do not show significant alterations.

In previous studies, it was shown that the sustained and reversible inhibition of mechanical hyperalgesia without causing sedation or reduced motor performance, the 24-h antinociceptive activity, and the maximum analgesia with doses that do not induce potential side effects, mean that the use of CTK 01512-2 in a clinical setting is possible [4,5]. This assertion agrees with the findings of this study since the administration of the same toxin dose (200 pmol/site) in rats did not show significant acute toxic effects. However, further studies are required to evaluate the repeated dose toxicity and toxicity in other species.

## 4. Conclusions

The preclinical signs presented after the intrathecal administration of the Phα1β and CTK 01512-2 toxins from the *Phoneutria nigriventer* spider to Wistar rats demonstrated that these toxins presented low acute toxicity. Thus, we conclude that the native Phα1β toxin and the recombinant CTK 01512-2 present a promise safety profile based on acute toxicity.

## 5. Materials and Methods

### 5.1. Drugs

Native Phα1β purified from *Phoneutria nigriventer* venom (FUNED, Belo Horizonte, Brazil); CTK 01512-2 (Giotto Biotech S.r.l, Florence, Italy); and MVIIA (Latoxan, Valence, France).

### 5.2. Animals

Adult male (*n* = 30) and female (*n* = 30) Wistar rats (60 old days) from ULBRA/Canoas bioterium were used. The protocol was performed after the University Ethics Committee approval (Number 2014-31P-Canoas) and was carried out in accordance with current guidelines for the care of laboratory animals. The animals kept in polypropylene boxes (41 × 34 × 16 cm) (4 rats per box) with free access to water and food in light/dark cycles of 12 h, in an environment with controlled temperature (22 ± 2 °C) and monitored humidity. All experiments were approved by the Animal Care and Use Committee of the Lutheran University of Brazil, Canoas/RS, Brazil (identification code: Protocol no: 2014-31P; date of approval: 17 November 2014.)

### 5.3. Acute Toxicity Protocol

The test was based on the fixed dose acute toxicity method (OECD 420). Both genders were included to perform the biochemical and histological evaluation in males, 24 h after exposure, as well as to follow the possible reversibility of the signs in females (slightly more sensitive), observed after 14 days (official protocol). Groups of 5 animals were treated through intrathecal administration with PBS 10 µL/site; MVIIA 200 pmol/site; CTK 01512-2 200 pmol (T1), 500 pmol/site (T2), and 1000 pmol/site (T3), and native Phα1β toxin 500 pmol/site. Each animal was observed for 1 min in the periods of 0, 15, 30, 60, 120, 240, and 360 min. Behaviors indicative of depression or stimulation of central nervous system activity and autonomic manifestations were observed. Signs indicative of toxicity observed were: locomotion alteration, reaction to stimuli, piloerection, diarrhea, sialorrhea, tremor, ptosis, altered muscle tone, hypnosis, convulsions, and abdominal contortions.

Males were euthanized 24 h and females 14 days after treatment. The males were anesthetized with isoflurane, the abdomen was incised, and the blood was collected from the caudal cava vein; organs (liver, kidneys, lungs, and spleen) were observed and removed to weight evaluation. Females were euthanized under isoflurane anesthesia and necropsied; organs (liver, kidneys, lungs, and spleen) were removed for weight evaluation. Clinical signs, macroscopic alterations, and relative organ weight (organ weight-related body weight) were evaluated in both genus; serum biochemistry and histopathology were evaluated in males and the females were observed daily for clinical signs and body weight for 14 days.

### 5.4. Biochemical Analysis

The blood samples were transferred to EDTA tubes and centrifuged at 1500× *g* for 10 min; the supernatant plasma was removed and stored in Eppendorf at −20 °C until analysis. Urea, creatinine, creatine kinase (CK), aspartate aminotransferase (AST), alanine aminotransferase (ALT), alkaline phosphatase (ALP), and amylase were tested by chemistry analyzer (BS120 – Mindray ^®^) through the enzymatic-colorimetric reactions using commercial kits. 

### 5.5. Histopathological Study

Male livers and kidneys were conserved in 10% buffered formalin for posterior histopathological analysis. After fixation, liver and kidney samples were processed and the paraffin blocks were sectioned in fragments of three µm. The fragments were mounted on slides, hematoxylin, and eosin stained (HE) and observed by an optical microscope at 40× and 400×.

### 5.6. Statistical Analysis

The results, expressed as a percentage of animals that presented the behavior or histological alteration (100% = 5 animals), were analyzed by chi-square test. The parametric data with normal distribution were evaluated through ANOVA (one-way or repeated measure) followed by Bonferroni test (relative organ or body weight and biochemical parameters). Non-parametric data were analyzed by Kruskal–Wallis (relative organ weight). A significance level of α = 0.05 (*p*-value < 0.05) was considered. All the analyses were performed by the Statistical Package for the Social Sciences (SPSS 21).

## Figures and Tables

**Figure 1 toxins-10-00531-f001:**
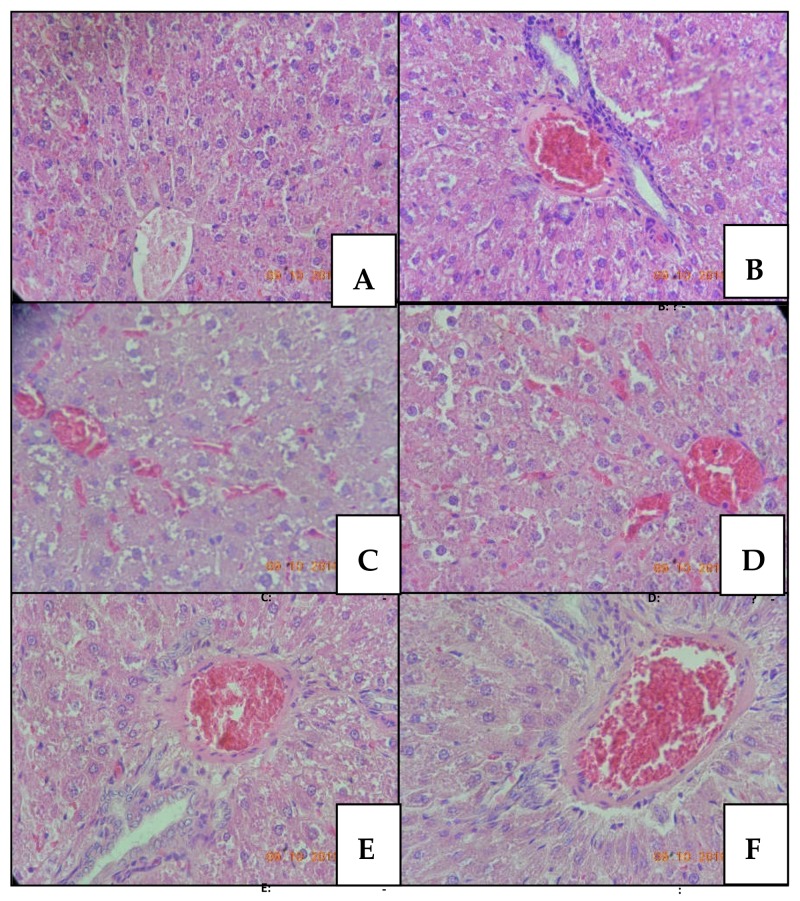
Liver tissue of male rats stained with hematoxylin-eosin and observed under optical microscope at 400× magnification. Rats treated with: PBS (**A**); MVIIA (**B**); CTK01512-2 T1 (**C**), T2 (**D**), T3 (**E**), and native Phα1β toxin (**F**).

**Figure 2 toxins-10-00531-f002:**
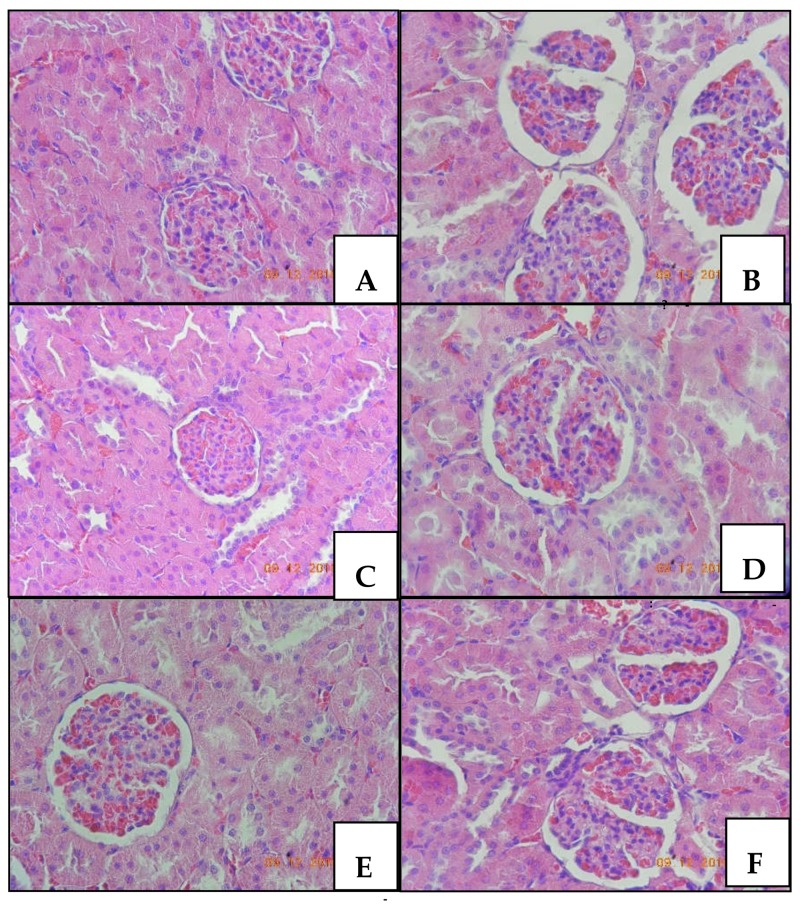
Renal tissue of male rats stained with hematoxylin-eosin and observed under optical microscope at 400× magnification. Rats treated with: PBS (**A**); MVIIA (**B**); CTK 01512-2 T1 (**C**), T2 (**D**), T3 (**E**), and native Phα1β toxin (**F**).

**Figure 3 toxins-10-00531-f003:**
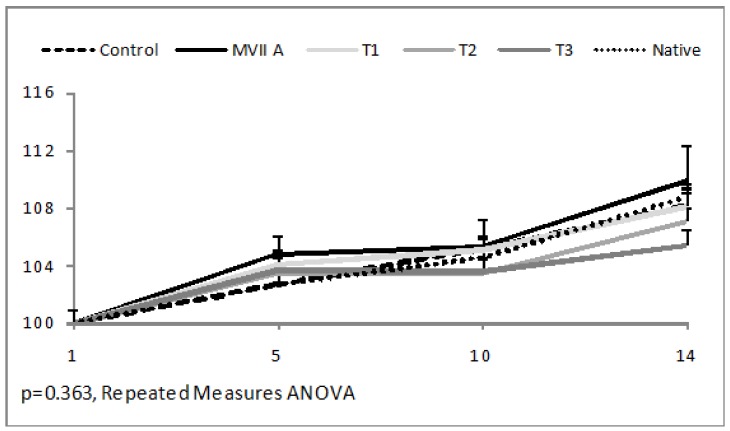
Relative body weight (%) of female rats after acute intrathecal administration native Phα1β toxin, T1, T2, or T3 of CTK 01512-2, PBS, and MVIIA during the evaluation period (14 days). Data are shown considering the first day body weight as 100% of five animals per group.

**Table 1 toxins-10-00531-t001:** Acute effects of native Phα1β toxin, T1, T2, and T3 of CTK 01512-2, PBS, or MVIIA after intrathecal administration to male rats. Results expressed as mean and standard error mean of five animals per group.

Toxic Signs (Occurrence Interval)	Proportion of Males that Express Signs (%) *n* = 5/Group					
	PBS 10 µL/Site	MVIIA 200 pmol/Site	CTK 01512-2 T1 200 pmol/Site	CTK 01512-2 T2 500 pmol/Site	CTK 01512-2 T3 1000 pmol/Site	Phα1β Native 500 pmol/Site
**Increased Ambulation (min)**	100 (0–15)	100 (0–15)	100 (0–15)	100 (0–15)	100 (0–15)	100 (0–15)
**Grooming** **(min)**	60 (0–60)	60 (0–120)	80 (0–360)	100 (0–360)	100 (0–120)	100 (0–120)
**Decreased Ambulation (min)**	100(15–360)	100 (15–360)	100 (15–360)	100 (15–360)	100 (15–360)	100 (15–360)
**Hypnosis *** **(min)**	80 ^a^ (120–360)	60 ^a^ (60–360)	60 ^a^ 120–360)	100 ^a^ (60–360)	60 ^a^ (120–360)	0 ^b^ (-)
**Decreased Touch Response (min)**	20 (60–120)	40 (15–360)	40 (15–360)	0 (-)	20 (15–240)	40 (15–360)
**Ataxia** **(min)**	0 (-)	20 (120–360)	0 (-)	0 (-)	0 (-)	0 (-)
**Piloerection** **(min)**	20 (240–360)	40 (15–360)	40 (30–360)	40 (60–360)	20 (30–240)	100 (15–360)
**Dyspnea *** **(min)**	0 ^a^ (-)	80 ^b^ (15–360)	80 ^b^ (15–360)	100 ^b^ (15–360)	100 ^b^ (15–360)	80 ^b^ (30–360)
**Tremors** **(min)**	0 (-)	20 (15, 360)	0 (-)	0 (-)	0 (-)	0 (-)
**Straub tail** **(min)**	0 (-)	20 (15–360)	0 (-)	0 (-)	0 (-)	20 (15–60)

** p* < 0.05 (Chi–Square); differences between groups are represented by letters (^a^ ≠ ^b^).

**Table 2 toxins-10-00531-t002:** Relative organ weight (%) of male rats, 24 h after acute intrathecal administration native Phα1β toxin, T1, T2, or T3 of CTK 01512-2, PBS, or MVIIA. Results expressed as mean and standard error mean of five animals per group.

Relative Organ Weight (%)	PBS 10 µL/Site	MVIIA 200 pmol/Site	CTK 01512-2 T1 200 pmol/Site	CTK 01512-2 T2 500 pmol/Site	CTK 01512-2 T3 1000 pmol/Site	Phα1β Native 500 pmol/Site
**Liver**	4.79 ± 0.18	4.70 ± 0.06	4.57 ± 0.17	4.91 ± 0.11	4.76 ± 0.22	4.63 ± 0.09
**Kidneys**	0.80 ± 0.02	0.77 ± 0.03	0.79 ± 0.02	0.81 ± 0.01	0.80 ± 0.02	0.74 ± 0.04
**Adrenal glands**	0.017 ± 0.001	0.015 ± 0.001	0.016 ± 0.001	0.016 ± 0.001	0.017 ± 0.001	0.014 ± 0.002
**Heart**	0.31 ± 0.01	0.32 ± 0.02	0.32 ± 0.001	0.32 ± 0.01	0.32 ± 0.01	0.31 ± 0.001
**Lung**	0.38 ± 0.01	0.41 ± 0.02	0.43 ± 0.02	0.42 ± 0.02	0.43 ± 0.02	0.39 ± 0.02
**Spleen**	0.24 ± 0.01	0.24 ± 0.01	0.25 ± 0.01	0.24 ± 0.01	0.24 ± 0.01	0.23 ± 0.1

*p* > 0.05 (ANOVA or Kruskal–Wallis).

**Table 3 toxins-10-00531-t003:** Biochemical parameters of male rats, 24 h after acute intrathecal administration of native Phα1β toxin, T1, T2, T3 of CTK 01512-2, PBS, and MVIIA. Results expressed as mean and standard error mean of five animals per group. * *p* < 0.05 (one-way ANOVA, Bonferroni); differences between groups are represented by letters. AST (aspartate aminotransferase); ALT (alanine aminotransferase); AL*P* (alkaline phosphatase); (CK) creatine kinase.

Biochemical Parameters	PBS 10 µL/Site	MVIIA 200 pmol/Site	CTK1512-2 T1 200 pmol/Site	CTK1512-2 T2 500 pmol/Site	CTK1512-2 T3 1000 pmol/Site	Phα1β Native 500 pmol/Site
**Urea *** **(mg/dL)**	58.0 ± 4.52 ^a^	61.0 ± 2.38 ^a^	55.2 ± 2.65 ^a^	45.2 ± 1.93 ^b^	47.8 ± 1.77 ^b^	44.7 ± 5.80 ^b^
**Creatinine** **(mg/dL)**	0.59 ± 0.032	0.57 ± 0.032	0.54 ± 0.02	0.58 ± 0.027	0.55 ± 0.018	0.48 ± 0.025
**AST *** **(U/L)**	192.8 ± 22.55 ^a^	210.0 ± 35.55 ^a^	191.2 ± 39.16 ^a^	145.2 ± 12.54 ^b^	146.2 ± 20.30 ^a,b^	109.7 ± 5.89 ^b^
**ALT*** **(U/L)**	107.0 ± 15.32 ^a^	97.1 ± 12.09 ^a^	94.6 ± 7.73 ^a^	80.0 ± 3.37 ^b^	77.2 ± 3.24 ^b^	66.0 ± 2.51 ^b^
**ALP *** **(U/L)**	244.2 ± 14.94 ^a^	201.3 ± 8.78 ^b^	292.2 ± 17.33 ^a^	250.8 ± 32.43 ^a^	281.0 ± 65.85 ^a^	177.7 ± 21.12 ^b^
**Amylase** **(U/L)**	1381.0 ± 201.42	1148.6 ± 83.12	1159.8 ± 119.92	1029.2 ± 77.81	968.4 ± 53.64	958.2 ± 69.34
**CK** **(U/L)**	388.2 ± 78.51	225.5 ± 18.06	378.6 ± 70.19	265.8 ± 36.82	340.6 ± 63.30	282.7 ± 40.36

* *p* < 0.05 (Chi–Square); differences between groups are represented by letters (^a^ ≠ ^b^).

**Table 4 toxins-10-00531-t004:** Acute effects of native Phα1β toxin, T1, T2, or T3 of CTK 01512-2, PBS, and MVIIA after intrathecal administration to female rats.

Toxic Signs (Occurrence Interval)	Proportion of females that express the signs (%) *n* = 5/group					
	PBS 10 µL/Site	MVIIA 200 pmol/Site	CTK 01512-2 T1 200 pmol/Site	CTK 01512-2 T2 500 pmol/Site	CTK 01512-2 T3 1000 pmol/Site	Phα1β Native 500 pmol/Site
**Increased Ambulation** **(min)**	100 (0–15)	100 (0–15)	100 (0–15)	100 (0–15)	100 (0–15)	100 (0–15)
**Grooming** **(min)**	60 (0–360)	60 (0–30)	100 (0–360)	60 (0–30)	100 (0–360)	60 (0–15)
**Decreased Ambulation** **(min)**	100 (15–360)	100 (15–360)	100 (15–360)	100 (15–360)	100 (15–360)	100 (15–240)
**Hypnosis** **(min)**	60 (240–360)	40 (240–360)	40 (240–360)	40 (30–240)	40 (60–360)	0 (-)
**Decreased Touch Response * (min)**	20 ^a^ (120–360)	100 ^b^ (30–360)	40 ^a^ (15–120)	100 ^b^ (30–360)	60 ^a^ (30–360)	100 ^b^ (15–360)
**Piloerection** **(min)**	80 (0–15)	80 (15–360)	100 (15–360)	100 (0–360)	40 (30–360)	80 (0–360)
**Dyspnea** **(min)**	80 (0–15)	80 (15–360)	100 (15–360)	100 (30–360)	80 (120–360)	100(30–360)
**Tremors *** **(min)**	0 ^a^ (-)	100^b^ (15, 360)	40 ^b^ (15–360)	80 ^b^ (30–360)	40 ^b^ (240–300)	60 ^b^ (30–360)
**Straub tail** **(min)**	0 (-)	40 (15–360)	0 (-)	0 (-)	20 (300–360)	0 (-)

* *p* < 0.05 (Chi–Square); differences between groups are represented by letters (^a^ ≠ ^b^).

**Table 5 toxins-10-00531-t005:** Relative organ weight (%) of female rats 14 days after acute intrathecal administration of native Phα1β toxin, T1, T2, or T3 of CTK 01512-2, PBS, and MVIIA. Results expressed as mean and standard error mean of five animals per group.

Relative Organ Weight (%)	PBS 10 µL/Site	MVIIA 200 pmol/Site	CTK 01512-2 T1 200 pmol/Site	CTK 01512-2 T2 500 pmol/Site	CTK 01512-2 T3 1000 pmol/Site	Phα1β Native 500 pmol/Site
**Liver**	4.29 ± 0.13	4.61 ± 0.21	4.38 ± 0.14	4.23 ± 0.23	4.45 ± 0.16	4.23 ± 0.13
**Kidneys**	0.82 ± 0.03	0.82 ± 0.02	0.81 ± 0.03	0.83 ± 0.05	0.83 ± 0.02	0.83 ± 0.03
**Adrenal glans ***	0.033 ± 0.002 ^a.b^	0.034 ± 0.002 ^a,b^	0.028 ± 0.002 ^a^	0.037 ± 0.001 ^b^	0.035 ± 0.001 ^a,b^	0.037 ± 0.001 ^b^
**Heart**	0.34 ± 0.01	0.35 ± 0.01	0.34 ± 0.01	0.32 ± 0.02	0.34 ± 0.01	0.32 ± 0.02
**Lung**	0.51 ± 0.03	0.50 ± 0.01	0.52 ± 0.05	0.50 ± 0.04	0.55 ± 0.02	0.50 ± 0.02
**Spleen ***	0.28 ± 0.01^a^	0.27 ± 0.01 ^a,b^	0.26 ± 0.01 ^a,b^	0.25 ± 0.02 ^a,b^	0.27 ± 0.01^a,b^	0.25 ± 0.01^b^

* *p* < 0.05 (one-way ANOVA, Bonferroni); differences between groups are represented by letters (^a^ ≠ ^b^).

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
