# Peer review of "Acute Toxicity of the Recombinant and Native Phα1β Toxin: New Analgesic from Phoneutria nigriventer Spider Venom"

_toxins, 2018, doi:10.3390/toxins10120531_

Round 1
Reviewer 1 Report
The manuscript entitled "Acute toxicity of the recombinant and native Phα1β 2 toxin: new analgesic from Phoneutria nigriventer spider poison" described the safety of the intrathecal acute exposure to Phα1β native toxin and CTK 1512-2, using animal model. The manuscript is well written.
I have a few suggestions to improve the impact and rigor of this paper.
(1) Overall:
Please use subscribe for Cav.
(2) Overall:
Please use italic for Phoneutria nigriventer in the main body of the manuscript.
(3) Page 1, lines 13-14:
I could not understand the following sentence easily:
“but both were appropriate to the species”.
Does this sentence mean that the decrease in weight of the adrenal gland and spleen is within the normal range? If so, please rewrite this sentence.
(4) Page 7, lines 153-154: It seems to me that the spleen of the native toxin treated group was smaller than the control group treated with PBS.
Author Response
Reviewer 1:
I have a few suggestions to improve the impact and rigor of this paper.
(1) Overall:
Please use subscribe for Cav.
We made as changes
(2) Overall:
Please use italic for Phoneutria nigriventer in the main body of the manuscript.
We made as changes
(3) Page 1, lines 13-14:
I could not understand the following sentence easily:
“but both were appropriate to the species”.
Does this sentence mean that the decrease in weight of the adrenal gland and spleen is within the normal range? If so, please rewrite this sentence.
The sentence was rewrite
"although significant differences in the female relative weight of the adrenal glands and spleen have been observed, these values are within the normal range."
(4) Page 7, lines 153-154: It seems to me that the spleen of the Native toxin treated group was smaller than the control group treated with PBS.
The mistake was adjusted to "smaller than".

Reviewer 2 Report
The article is very interesting, well written and results are well presented. However, one major problem in the article is the fact that the authors treated the two groups of objects (males and females) differently. As far as I understood why there is no analysis of relative body weight changes in males, I cannot explain why there is no biochemical and histopathological analyses of female group. Especially since these tests showed significant differences in the males group. If the authors cannot explain that differences convincingly I recommend major revision.
Technical notes:
All binomial Latin names should be written in italic and the genus names should start with a capital letter – lines 20, 23, 31,199, 212
Table 3 “Urea” not “”Ureia”
Line 107 Delete “Hydropic degeneration (A)”
Author Response
Reviewer 2:
The article is very interesting, well written and results are well presented. However, one major problem in the article is the fact that the authors treated the two groups of objects (males and females) differently. As far as I understood why there is no analysis of relative body weight changes in males, I cannot explain why there is no biochemical and histopathological analyses of female group. Especially since these tests showed significant differences in the males group. If the authors cannot explain that differences convincingly I recommend major revision.
As described in the methodology the acute toxicity protocol of OECD 420 provides for monitoring for 14 days after single dose administration in order to verify the possible reversibility of the signs of toxicity presented by the substance. However, the authors consider that in order to evaluate possible biochemical and histopathological alterations of the acute exposure it is necessary to perform the analysis in 24 hours. Therefore, it was decided to perform this modification in the protocol in males, maintaining the original protocol in females (slightly more sensitive), ensuring both evaluations, acute biochemical and histopathological changes and possible reversibility of clinical signs and body mass.
In the text the sentence was changed to " The test was based on the fixed dose acute toxicity method (OECD 420). Both genders were included to perform the biochemical and histological evaluation in male, 24 hours after exposure, as well as follow the possible reversibility of the signs in female (slightly more sensitive), observed after 14 days (official protocol)."
Technical notes:
All binomial Latin names should be written in italic and the genus names should start with a capital letter – lines 20, 23, 31,199, 212
Table 3 “Urea” not “”Ureia”
We made as changes
Line 107 Delete “Hydropic degeneration (A)”
We made as changes

Round 2
Reviewer 2 Report
The explanation is fully satisfactory. Thank you for claryfying the contents of the manuscript